# Disproportionate use of polysubstance combinations varies by sexual identity among US adults

Luis M. Mestre[1]*, Marney A. White[1,2], Juhan Lee[3], Maria A. Parker[4], Krysten W. Bold[1]

1 Department of Psychiatry, Yale School of Medicine, New Haven, Connecticut, United States of America, 2 Department of Social and Behavioral Sciences, Yale School of Public Health, New Haven, Connecticut, United States of America, 3 Department of Epidemiology & Population Health, Montefiore Einstein, Albert Einstein College of Medicine, New York City, New York, United States of America, 4 Indiana University School of Public Health – Bloomington, Department of Epidemiology and Biostatistics, Bloomington, Indiana, United States of America

* luis.mestre@yale.edu

## Abstract

Polysubstance use is a major public health concern affecting lesbian, gay, and bisexual (LGB) adults, especially bisexual female adults. This study aims to identify the commonly used polysubstance combinations by LGB adults in the past 30 days and to determine whether these combinations differ by sexual identity and sex. Our analytic sample consisted of NSDUH 2021 and 2022 (n = 66,634 adults; 8.59% LGB adults). We used survey-weighted multinomial logistic regression models to assess the polysubstance combinations by sex. The most used substances were binge alcohol drinking, cannabis, cigarettes, and nicotine vape. Bisexual female adults used most of the assessed polysubstance combinations that included binge alcohol drinking, cannabis, or both, often involving three or four substances. Sex differences among the polysubstance combinations vary among heterosexual and bisexual adults but not among gay/lesbian adults. Public health strategies must consider the specific sexual identity, sex, and the types of substance combinations involved.

## Introduction

Polysubstance use, the use of multiple substances, is a public health concern. Polysubstance use increases the risk of chronic diseases and mental health illnesses like depression and anxiety disorders, along with increased rates of preventable deaths [1–5]. Lesbian, gay, and bisexual (LGB) adults have a higher prevalence of substance use than heterosexual adults [6–15]. These differences are partly due to LGB adults engaging in substance use to cope with the social stress they experience [10,16–19]. Bisexual adults face more social stress than gay or lesbian adults because of bi-erasure and biphobia, which leads to discrimination [20–25]. Facing more social stress is especially common among bisexual female adults [20,21,25,26],

**Data availability statement:** The data underlying this study are the National Survey on Drug Use and Health (NSDUH) 2021 and 2022 data sets available in the following database: https://www.samhsa.gov/data/data-we-collect/nsduh-national-survey-drug-use-and-health/datafiles.

**Funding:** This study was financially supported by the Intramural Research Program, National Institute on Drug Abuse in the form of a grant awarded to LMM (5T32DA019426-19). The funders had no role in study design, data collection and analysis, decision to publish, or preparation of the manuscript. No additional external funding was received for this study.

**Competing interests:** The authors have declared no competing interests exist.

who are more likely to use polysubstance combinations compared by sexual identity and sex [12,14,27].

Multiple studies have examined polysubstance use among LGB adults in the US, focusing on sex differences [9,10,12,14,27–33]. However, although these studies provide important insights, they have some limitations. A major limitation is that they do not use a representative sample of the US population. Many of these studies focus primarily on young adults [9,14,27,30–33] or LGB individuals from specific locations, like cities, community programs, or college campuses, which limits the external validity of the findings to all LGB adults in the US [9,14,27,28,32,33].

We identified two research gaps: (a) Not identifying the most used polysubstance combinations among LGB adults in the US. (b) Whether there are sex differences among LGB adults in the US regarding these polysubstance combinations. If such differences exist, we also aim to determine whether they vary by sexual identity among LGB adults nationwide. Addressing these research gaps can help inform clinical and public health initiatives, enabling us to customize substance use treatments and improve access for LGB adults based on their sex [34].

We hypothesized that sex differences in the polysubstance combinations will vary by sexual identity. We also hypothesized that bisexual female adults will have higher rates among all polysubstance combinations compared to all the other groups. We used data from the National Survey on Drug Use and Health (NSDUH) – a cross sectional study representative of the civilian, non-institutionalized US population, which is often used to assess polysubstance use among US adults [8,11–13]. Our representative sample includes LGB adults, our population of interest.

## Methods

### Study sample

The analytic sample from 2021 and 2022 of the NSDUH dataset was n = 66,634 adults, with 8.52% identifying as LGB [35]. The survey design changed in 2021, and earlier survey cycles had different survey collection procedures. Before 2021, NSDUH's staff collected the data only in person, while since then, these data have been collected both in person and online [35]. We only included US adults using a complete case approach (i.e., with valid responses) in our analysis. This study followed Strengthening the Reporting of Observational Studies in Epidemiology (STROBE) guidelines [36].

**Ethical statement.** Ethical approval was not required. This study was waived from the Institutional Review Board (IRB) of the corresponding author's institution oversight due to the nature of secondary analyses of publicly available, de-identified datasets. The data used in this study is secondary data. The IRB waived the need for ethics approval and patient consent for the collection, analysis, and publication of the de-identified data for this secondary data analysis.

### Measures

**Independent variable (exposure) – Sexual identity.** We used sexual identity as the independent variable in this study. The survey assessed sexual identity using

the question, *"Which of the following do you consider yourself to be?"* Consistent with other studies, the participants self-identified either as heterosexual, gay/lesbian, bisexual, or do not know (i.e., Not Sure) [8,11,12].

**Dependent variable (outcome) – Past 30 days used polysubstance combinations.** We defined polysubstance use as the different combinations of the four most used substances in the past 30 days: Binge Alcohol Drinking, Cannabis, Cigarettes, and Nicotine Vape – S1 Table. We used this approach to build upon earlier studies that looked at only two or three substances [14,17,37], as considering more options gives us better insight into the complex behavior of polysubstance use [1,4].

We used the imputed variables of past 30-day substances available in NSDUH 2021 and 2022 (S1 Table). Substances included binge alcohol drinking – five or more drinks at the same time or within a couple of hours of each other for male and four or more drinks for female adults; cannabis; cigarettes; cocaine; crack; hallucinogens (such as foxy, LSD – acid, ecstasy – molly, ketamine, PCP – angel dust, peyote, mescaline, psilocybin, dimethyltryptamine (i.e., DMT), alpha-methyltryptamine (i.e., AMT), and salvia divinorum); heroin; inhalants; methamphetamine; cannabis vape, flavor vape, and nicotine vape. Misused substances included: prescription pain relievers (including fentanyl); sedatives; stimulants, and tranquilizers.

Based on the four most used substances, we had 12 mutually exclusive combinations. The combinations comprised two, three, or four most used substances (**Table 1**). The mutually exclusive combinations were: (a) No use of any of the four most used substances – Binge Alcohol Drinking, Cannabis, Cigarettes, or Nicotine Vape (the reference group of this study). (b) Binge Alcohol Drinking + Cannabis. (c) Binge Alcohol Drinking + Cannabis + Cigarettes. (d) Binge Alcohol Drinking + Cannabis + Cigarettes + Nicotine Vape. (e) Binge Alcohol Drinking + Cannabis + Nicotine Vape. (f) Binge Alcohol Drinking + Cigarettes. (g) Binge Alcohol Drinking + Cigarettes + Nicotine Vape. (h) Binge Alcohol Drinking + Nicotine Vape. (i) Cannabis + Cigarettes. (j) Cannabis + Cigarettes + Nicotine Vape. (k) Cannabis + Nicotine Vape. (l) Cigarettes + Nicotine Vape.

**Covariates.** We included other variables related to sexual identity and polysubstance use as done in previous studies [6,12,13,38–40]. These variables included age, survey cycle, race/ethnicity, sex, educational attainment, employment status, urbanicity, household income, marital status, and medical insurance status. **Table 1** outlines how we operationalized these variables.

## Statistical analysis

We used a complete cases approach to handle missing data, which removed 6,028 participants (6.39% of our sample). This small percentage of missing data is unlikely to bias our results [41]. We implemented NSDUH 2021 and 2022 survey weights; we divided the survey weights by two, the number of survey cycles included in the study [35]. We conducted a survey-weighted Adjusted Wald Chi-Square test to examine whether: (a) there were sex differences among all poly-substance combinations by sexual identity. (b) Associations between sexual identity and other variables. We also did survey-weighted t-tests to compare the estimated prevalence of: (a) the polysubstance combinations; (b) the substances included in the study to determine the polysubstance combinations; and (c) the male and female adults within sexual identity by each polysubstance combination.

We used a survey-weighted multinomial model to examine the association between sexual identity and the polysubstance combinations of the four most used substances – binge alcohol drinking, cannabis, cigarettes, and nicotine vape. Our model had "no use of any of the following – Binge Alcohol Drinking, Cannabis, Cigarettes, or Nicotine Vape" as the reference group, consistent with the reference for other studies [9,14]. The polysubstance combinations of the four most used substances, our outcome, consisted of 12 mutually exclusive categories. The multinomial model was adjusted by the covariates. We also included an interaction term of sexual identity and sex to compare female adults (lesbian, bisexual, and heterosexual) with male adults (gay, bisexual, and heterosexual) to determine whether there are sex differences by sexual identity. The independent variable of the multinomial model was the interaction between sexual identity and sex. We also did additional multinomial models with the same

**Table 1. Descriptive Statistics by sexual identity, NSDUH 2021 and NSDUH 2022.**

| Variables | n (weighted %[a]; SE); p-value | | | | | |
| --- | --- | --- | --- | --- | --- | --- |
| | Gay/Lesbian | Bisexual | Heterosexual | Not Sure[d] | Total | |
| | 1,889 (2.44) | 5,886 (5.50) | 58,465 (91.48) | 394 (0.58) | 66,634 | p-value[b] |
| **Polysubstance Combinations** | | | | | | |
| No use of any of the following – Binge Alcohol Drinking, Cannabis, Cigarettes, or Nicotine Vape | 1,194 (66.67; 1.89); p<0.001*,c, e | 3,173 (53.15; 1.26); p<0.001*,c, e | 45,789 (81.00; 0.27) | 340 (91.41; 1.82) | 50,496 (79.18) | F(18, 33) = 18.80; p<0.001* |
| Binge Alcohol Drinking+Cannabis | 168 (9.18; 0.88); p<0.001*,c, e | 570 (9.47; 0.81); p<0.001*,c, e | 2,761 (3.84; 0.14) | 8 (0.82; 0.39) | 3,507 (4.27) | |
| Binge Alcohol Drinking+Cannabis+Cigarettes | 69 (3.19; 0.48); p<0.001*,c, e | 268 (5.82; 0.48); p<0.001*,c, e | 1,371 (2.31; 0.11) | 5 (0.35; 0.18) | 1,713 (2.52) | |
| Binge Alcohol Drinking+Cannabis+Cigarettes+Nicotine vape | 49 (1.59; 0.35); p<0.001*,c, e | 242 (3.52; 0.40); p<0.001*,c, e | 692 (0.76; 0.04) | 2 (0.03; 0.02) | 985 (0.93) | |
| Binge Alcohol Drinking+Cannabis+Nicotine vape | 59 (2.27; 0.87); p=0.004*,c; | 311 (5.23; 0.43); p<0.001*,c, e | 935 (1.01; 0.06) | 3 (0.60; 0.45) | 1,308 (1.27) | |
| Binge Alcohol Drinking+Cigarettes | 73 (4.21; 0.76); p<0.001*,c, e | 187 (4.17; 0.58); p<0.001*,c, e | 2,048 (4.00; 0.15) | 11 (1.45; 0.74) | 2,319 (4.00) | |
| Binge Alcohol Drinking+Cigarettes+Nicotine vape | 30 (0.99; 0.25); p<0.001*,c, e | 137 (1.90; 0.28); p<0.001*,c, e | 540 (0.72; 0.05) | 4 (1.10; 0.79) | 711 (0.79) | |
| Binge Alcohol Drinking+Nicotine vape | 46 (1.39; 0.34); p<0.001*,c, e | 191 (2.51; 0.28); p<0.001*,c, e | 1,132 (1.29; 0.07) | 4 (0.24; 0.14) | 1,373 (1.35) | |
| Cannabis+Cigarettes | 91 (5.64; 0.81); p<0.001*,c, e | 231 (4.91; 0.54); p<0.001*,c, ee | 1,403 (2.57; 0.15) | 9 (1.92; 1.15) | 1,734 (2.77) | |
| Cannabis+Cigarettes+Nicotine vape | 34 (1.75; 0.50); p<0.001*,c, e | 151 (2.78; 0.36); p<0.001*,c, e | 424 (0.55; 0.05) | 2 (0.96; 0.68) | 611 (0.70) | |
| Cannabis+Nicotine vape | 48 (1.50; 0.35); p<0.001*,c, e | 285 (3.95; 0.41); p<0.001*,c, e | 750 (0.95; 0.07) | 5 (0.54; 0.30) | 1,088 (1.13) | |
| Cigarettes+Nicotine vape | 28 (1.61; 0.47); p<0.001*,c, e | 140 (2.58; 0.33); p<0.001*,c, e | 620 (0.99; 0.07) | 1 (0.58; 0.57) | 789 (1.09) | |
| **Age Group** | | | | | | |
| 18-29 years | 964 (28.93; 1.46) | 4,114 (55.29; 1.33) | 20,549 (17.71; 0.29) | 215 (28.80; 3.29) | 25,842 (20.11) | F(6, 45) = 106.46; p<0.001* |
| 30-49 years | 646 (36.86; 1.92) | 1,541 (32.67; 1.10) | 22,231 (31.66; 0.32) | 129 (37.98; 3.83) | 24,547 (31.88) | |
| ≥50 years | 279 (34.21; 2.27) | 231 (12.03; 1.26) | 15,685 (50.63; 0.46) | 50 (33.21; 4.96) | 16,245 (49.09) | |
| **Sex** | | | | | | |
| Male | 917 (40.67; 2.07) | 1,357 (29.05; 1.26) | 26,803 (48.17; 0.34) | 123 (34.66; 4.29) | 29,200 (47.31) | F(3, 48) = 69.70; p<0.001* |
| Female | 972 (59.33; 2.07) | 4,529 (70.95; 1.26) | 31,662 (51.83; 0.34) | 271 (65.34; 4.29) | 37,434 (52.69) | |
| **Race/Ethnicity** | | | | | | |
| Hispanic | 305 (16.84; 1.60) | 975 (16.51; 1.22) | 9,470 (16.86; 0.54) | 110 (26.23; 3.93) | 10,860 (16.90) | F(15, 36) = 4.36; p<0.001* |
| Non-Hispanic Asian | 89 (3.97; 1.03) | 231 (4.49; 0.41) | 3,763 (6.91; 0.31) | 63 (20.88; 3.64) | 4,146 (6.79) | |
| Non-Hispanic Black | 218 (13.11; 1.32) | 533 (10.49; 0.85) | 6,731 (12.23; 0.39) | 52 (13.98; 3.03) | 7,534 (12.17) | |
| Non-Hispanic More than One Race | 93 (2.87; 0.61) | 379 (3.53; 0.50) | 1,997 (1.70; 0.09) | 19 (1.00; 0.35) | 2,488 (1.82) | |
| Non-Hispanic Native/Alaskan Native/ Hawaiian Native | 36 (1.23; 0.61) | 89 (1.16; 0.33) | 847 (0.85; 0.07) | 13 (2.44; 0.98) | 985 (0.90) | |
| Non-Hispanic White | 1,148 (61.98; 1.95) | 3,679 (63.82; 1.38) | 35,657 (61.44; 0.74) | 137 (35.47; 4.25) | 40,621 (61.43) | |

*(Continued)*

| Variables | n (weighted %[a]; SE); p-value | | | | | |
|---|---|---|---|---|---|---|
| | Gay/Lesbian | Bisexual | Heterosexual | Not Sure[d] | Total | |
| **Survey Year** | | | | | | |
| 2021 | 916 (44.31; 2.09) | 2,670 (45.26; 1.27) | 29,260 (49.60; 0.52) | 185 (47.87) | **33,031 (49.22)** | $F_{(3,48)} = 4.51$; |
| 2022 | 973 (55.69; 2.09) | 3,216 (54.74; 1.27) | 29,205 (50.40; 0.52) | 209 (52.13) | **33,603 (50.78)** | $p = 0.007$* |
| **Educational Attainment** | | | | | | |
| Less than High School | 426 (6.40; 0.84) | 662 (10.11; 1.21) | 5,543 (10.11; 0.27) | 98 (26.57; 3.44) | **6,446 (10.11)** | $F_{(9,42)} = 13.09$; |
| High School Graduate | 143 (23.07; 1.88) | 1,609 (26.93; 1.26) | 13,986 (27.04; 0.42) | 129 (31.32; 3.50) | **16,150 (26.96)** | $p < 0.001$* |
| Some College/ Associate Degree | 600 (31.04; 1.88) | 2,118 (37.23; 0.81) | 16,948 (29.35; 0.39) | 94 (28.78; 3.85) | **19,760 (29.83)** | |
| College Graduate | 720 (39.49; 1.89) | 1,497 (25.73; 1.42) | 21,988 (33.50; 0.39) | 73 (13.33; 2.80) | **24,278 (33.10)** | |
| **Household Income** | | | | | | |
| < 20,000 USD | 860 (38.14; 2.14) | 3,547 (56.02; 1.44) | 23,287 (35.78; 0.49) | 280 (65.74; 4.50) | **27,974 (37.13)** | $F_{(3,48)} = 58.35$; |
| ≥ 20,000 USD | 1,029 (61.86; 2.14) | 2,339 (43.98; 1.44) | 35,178 (64.22; 0.49) | 114 (34.26; 4.50) | **38,660 (62.87)** | $p < 0.001$* |
| **Urbanicity** | | | | | | |
| Non-Metro | 233 (7.62; 0.51) | 937 (12.99; 1.14) | 9,308 (14.02; 0.81) | 59 (8.68; 1.96) | **10,537 (13.78)** | $F_{(3,48)} = 11.91$; |
| Metro | 1,656 (92.38; 0.51) | 4,949 (87.01; 1.14) | 49,157 (85.98; 0.81) | 335 (91.32; 1.96) | **56,097 (86.22)** | $p < 0.001$* |
| **Medical Insurance** | | | | | | |
| No | 196 (8.48; 1.43) | 712 (12.01; 0.86) | 5,552 (8.60; 0.29) | 76 (19.53; 2.89) | **6,536 (8.85)** | $F_{(3,48)} = 9.42$; |
| Yes | 1,693 (91.52; 1.43) | 5,174 (87.99; 0.86) | 52,913 (91.40; 0.29) | 318 (80.47; 2.89) | **60,098 (91.15)** | $P < 0.001$* |
| **Employment status** | | | | | | |
| No | 196 (6.62; 1.07) | 712 (8.67; 0.63) | 5,552 (4.20; 0.16) | 76 (5.25; 1.63) | **6,536 (8.85)** | $F_{(3,48)} = 15.26$; |
| Yes | 1,693 (93.38; 1.07) | 5,174 (91.33; 0.63) | 52,913 (95.80; 0.16) | 318 (94.75; 1.63) | **60,098 (91.15)** | $p < 0.001$* |
| **Marital Status** | | | | | | |
| No | 1,524 (75.99; 1.70) | 4,780 (73.31; 1.05) | 31,746 (47.99; 0.42) | 268 (56.51; 4.83) | **38,318 (50.39)** | $F_{(3, 48)} = 200.59$; |
| Yes | 365 (24.01; 1.70) | 1,106 (21.69; 1.05) | 26,719 (52.01; 0.42) | 126 (43.49; 4.83) | **28,316 (49.61)** | $p < 0.001$* |

[a] Some weighted percentages will not add to 100% due to rounding error [b] Survey-weighted Adjusted Wald Chi-Square Test.

"*" p-value < 0.05.

[c] Survey-Weighted t-test with 49 degrees of freedom (Reference group; heterosexual adults)

[d] Due to low statistical power, we did not compared the proportions of the substance combinations of the "Not Sure" group with heterosexual adults

[e] Bonferroni Correction (p < 0.004). Correction for the multiple comparisons of the survey-weighted t-tests (12 comparisons in total, one per category).

outcome, polysubstance combinations of the four most used substances, to compare lesbian and bisexual female adults with other groups such as gay male or bisexual male adults. The model diagnostics of each model are S2 in S1 Table.

As part of our sensitivity analyses, we implemented multiple imputations for sexual identity and medical insurance, variables that were not imputed by NSDUH. Then, we compared the imputed and observed distributions of these variables using a survey-weighted Rao-Scott Chi-Square goodness-of-fit test. We found no difference between the observed and imputed variables (S3 in S1 Table). We also did two sample proportions tests to determine whether there were differences in the estimated prevalence of the used substances ranked 4th, our threshold, and 5th by sexual identity. The family-wise error rate (i.e., the risk of false positives due to multiple testing) was 45.96% for 12 comparisons, 40.13% for ten, and 55.99% for 16 comparisons. Due to the risk of having at least one false positive was high in any of the of multiple testing we adjusted for multiple comparisons, we included a Bonferroni Correction in our sensitivity analyses for the survey-weighted t-tests and the multinomial models. The thresholds to determine a significant p-value with Bonferroni Corrections in these t-tests were 0.004, 0.005, or 0.003; 0.05 divided by the 12 mutually exclusive categories of the polysubstance combination variable, ten multinomial models to compare female adults by sexual identity with all the other sexual identity by sex groups, 16 for the substances that were considered to operationalize the polysubstance combinations variable, respectively. We did all the analyses in R Studio 3.16.

## Results

Table 1 shows the descriptive statistics of the analytic sample, including the survey-weighted prevalence, by sexual identity, of the polysubstance combinations of two, three, or four substances. Bisexual adults had the lowest prevalence of having medical insurance, employment, an income of at least 20,000 USD, and a college degree compared to heterosexual and gay/lesbian adults (Table 1).

The most used substances in the past 30 days among gay/lesbian and bisexual adults were binge alcohol drinking, cannabis, cigarettes, and nicotine vape (S1 Table). Bisexual adults had the highest prevalence of past 30-day use of these four most common substances compared to any other sexual identity (S1 Table). We also found significant differences between the use of nicotine vape and cannabis vape by sexual identity groups; the substances ranked 4th, our threshold, and 5th, respectively, among heterosexual ($\chi^2(1)=1,620,361$; $p<0.001$), gay/lesbian ($\chi^2(1)=130,384$; $p<0.001$), and bisexual adults ($\chi^2(1)=5,288.4$; $p<0.001$)). Therefore, we did not lose significant information by using a cut-off of 4 substances.

### Sex Differences in the Polysubstance Combinations vary by Sexual Identity

S4 in S1 Table shows the prevalence of these common polysubstance combinations among different sexual identities and sexes. We found that for all combinations there were sex differences within the sexual identities, heterosexual, gay/lesbian, and bisexual adults (S4 in S1 Table and Fig 1). Among heterosexual adults, male adults engaged more than female adults in most polysubstance combinations while among bisexual adults, female adults engaged more than male adults. Among gay/lesbian adults, although there were sex differences in all polysubstance combinations, they did not vary disproportionately more either among gay male or lesbian female adults.

We were able to confirm these sex differences patterns with the results from the Weighted Wald Adjusted Chi-Square Test. These findings indicate that sex was not associated with using polysubstance combinations among gay/lesbian adults (F(11,40) = 1.39; p = 0.216). However, sex was associated with these polysubstance combinations among heterosexual (F(11,40) = 17.29; p < 0.001) and bisexual adults (F(11,40) = 2.79; p < 0.001; S5 in S1 Table).

### Bisexual female adults have higher rates among polysubstance combinations compared to heterosexual female, heterosexual male, gay male, and lesbian female adults

Lesbian female adults engaged more in polysubstance combinations than heterosexual female adults (Table 2). Lesbian female adults only engaged more in one combination than heterosexual male adults though they engaged less in one combination compared to bisexual and gay male adults. Bisexual female adults led in the use of most polysubstance combinations

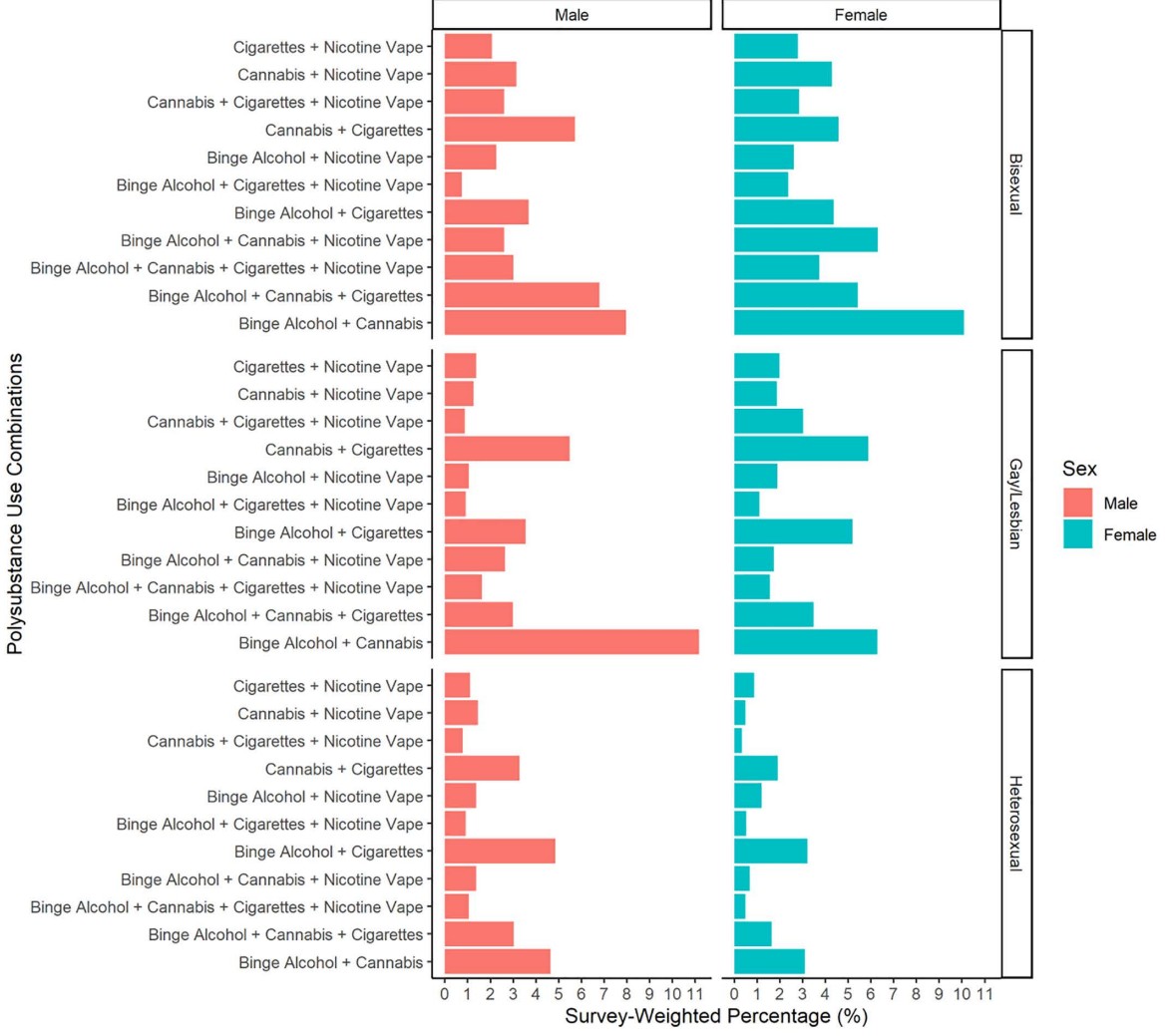

**Fig 1. Survey-weighted prevalence of polysubstance combinations by sexual identity and sex, NSDUH 2021-2022.**

compared to all female adult groups and heterosexual male adults, while heterosexual female adults used them the least compared to all groups (Tables 3–4, and Table 5). Bisexual female adults used more polysubstance combinations that include binge alcohol drinking, cannabis, or both, often involving three or four substances (Table 3 and Table 4).

In the same model, we found that adults from 30–49 years and ≥50 years engaged more than 18–29 years adults in substance combinations that include cigarettes (S6 in S1 Table). Gay/lesbian and bisexual adults were mainly in the 18–29 years and 30–49 years groups (Table 1), indicating that some of these polysubstance combinations that were frequent among LGB adults are due to them being younger than heterosexual adults. The full multinomial model of the analytic sample size (n = 66,634) is in S7 in S1 Table.

## Discussion

The key findings of this study are the following: (a) there were sex differences in polysubstance use among bisexual and heterosexual adults, as other studies have shown [9,10,12,14,28–31,33]; (b) bisexual female adults were at the highest

**Table 2. Summary of Survey Weighted Multinomial Models with potential combinations by sexual identity and sex among lesbian female adults, NSDUH 2021 and 2022.**

| Polysubstance Combinations | Lesbian female vs. Heterosexual Male [a] | | | Lesbian Female vs. Heterosexual Female [b] | | | Lesbian Female vs. Bisexual Male [c] | | | Lesbian Female vs. Gay Male [d] | | |
|---|---|---|---|---|---|---|---|---|---|---|---|---|
| | PR | 95% CI | p-value | PR | 95% CI | p-value | PR | 95% CI | p-value | PR | 95% CI | p-value |
| **Binge Alcohol Drinking + Cannabis** | 0.78 | (0.41, 1.49) | 0.452 | 1.54 | (0.96, 2.46) | 0.074 | 0.78 | (0.44, 1.85) | 0.507 | 0.45 | (0.41, 1.22) | 0.004*, e |
| **Binge Alcohol Drinking + Cannabis + Cigarettes** | 1.78 | (0.77, 3.98) | 0.157 | 1.94 | (1.11, 3.40) | 0.020* | 1.65 | (0.41, 3.74) | 0.369 | 0.81 | (0.38, 2.19) | 0.631 |
| **Binge Alcohol Drinking + Cannabis + Cigarettes + Nicotine vape** | 1.31 | (0.48, 3.56) | 0.594 | 1.91 | (1.02, 3.58) | 0.043* | 0.99 | (0.37, 2.71) | 0.984 | 0.61 | (0.48, 2.09) | 0.308 |
| **Binge Alcohol Drinking + Cannabis + Nicotine vape** | 0.73 | (0.18, 2.99) | 0.665 | 1.29 | (0.71, 2.34) | 0.411 | 0.26 | (0.19, 1.65) | 0.014* | 0.56 | (0.27, 2.25) | 0.280 |
| **Binge Alcohol Drinking + Cigarettes** | 2.01 | (0.88, 4.58) | 0.099 | 1.75 | (1.00, 3.05) | 0.049* | 1.3 | (0.29, 4.39) | 0.704 | 1.03 | (0.50, 2.03) | 0.935 |
| **Binge Alcohol Drinking + Cigarettes + Nicotine vape** | 1.51 | (0.53, 4.27) | 0.441 | 1.56 | (0.75, 3.26) | 0.236 | 0.33 | (0.05, 7.04) | 0.369 | 0.88 | (0.36, 2.48) | 0.790 |
| **Binge Alcohol Drinking + Nicotine vape** | 1.35 | (0.47, 3.89) | 0.582 | 0.99 | (0.47, 2.06) | 0.977 | 2.73 | (0.49, 4.92) | 0.087 | 1.55 | (0.44, 3.34) | 0.400 |
| **Cannabis + Cigarettes** | 1.77 | (0.61, 5.10) | 0.292 | 3.39 | (1.80, 6.39) | <0.001*, e | 1.55 | (0.16, 9.03) | 0.670 | 0.81 | (0.30, 2.83) | 0.722 |
| **Cannabis + Cigarettes + Nicotine vape** | 5.53 | (1.92, 15.94) | 0.002*,e | 7.06 | (3.26, 15.26) | <0.001*, e | 4.32 | (0.41, 8.65) | 0.059 | 2.74 | (0.62, 3.87) | 0.307 |
| **Cannabis + Nicotine vape** | 2.89 | (1.00, 8.33) | 0.048* | 2.44 | (1.11, 5.38) | 0.028* | 1.55 | (0.34, 4.37) | 0.500 | 1.01 | (0.34, 3.00) | 0.982 |
| **Cigarettes + Nicotine vape** | 1.65 | (0.36, 7.61) | 0.523 | 2.27 | (0.85, 6.09) | 0.104 | 1.36 | (0.12, 11.27) | 0.791 | 1.43 | (0.25, 5.40) | 0.649 |

"*" p-value < 0.05.

PR = Prevalence Ratio.

CI = Confidence Interval.

NOTE: These are mutually exclusive categories. The reference group of the multinomial models was "No use".

[a]The sample of this model was n = 66,634. We adjusted the model by sexual identity, age group, sex, race/ethnicity, survey cycle, educational attainment, household income, urbanicity, medical insurance, employment status, and marital status. We also included an interaction term between sexual identity and sex on this model. The independent variable of this model was the interaction between sexual identity and sex.

[b]Only female adults were included in this analysis. The subsample for this model was n = 37,434. We adjusted the model by sexual identity, age group, race/ethnicity, survey cycle, educational attainment, household income, urbanicity, medical insurance, employment status, and marital status. The independent variable of this model was sexual identity.

[c]Only LGB adults were included in this analysis. The subsample for this model was n = 8,169. We adjusted the model by sexual identity, age group, sex, race/ethnicity, survey cycle, educational attainment, household income, urbanicity, medical insurance, employment status, and marital status. We also included an interaction term between sexual identity and sex on this model. The independent variable of this model was the interaction between sexual identity and sex.

[d]Only gay/lesbian adults were included in this analysis. The subsample for this model was n = 1,889. We adjusted the model by age group, sex, race/ethnicity, survey cycle, educational attainment, household income, urbanicity, medical insurance, employment status, and marital status. The independent variable of this model was sex.

[e]Bonferroni Correction (p < 0.005). Correction for the multiple comparisons of using multiple multinomial models (10 multinomial models in total, one per model).

risk for engaging in most of the used polysubstance combinations compared to heterosexual female adults, who were at the lowest risk, as seen in other studies [8,9,12,14,27,32,33]. This pattern holds even when looking at heterosexual or gay male adults and heterosexual or lesbian female adults as the reference group. (c) We also found significant differences based on sex and sexual identity in most polysubstance combinations, which was not done in previous studies.

**Table 3. Summary of Survey Weighted Multinomial Models with potential combinations by sexual identity and sex among bisexual female adults versus male adults, NSDUH 2021 and 2022.**

| Polysubstance Combinations | Bisexual Female vs. Heterosexual Male [a] | | | Bisexual Female vs. Bisexual Male [b] | | | Bisexual Female vs. Gay Male [c] | | |
|---|---|---|---|---|---|---|---|---|---|
| | PR | 95% CI | p-value | PR | 95% CI | p-value | PR | 95% CI | p-value |
| Binge Alcohol Drinking+Cannabis | 2.04 | (1.38, 3.02) | <0.001*, e | 0.97 | (0.63, 1.97) | 0.920 | 1.28 | (0.54, 2.29) | 0.507 |
| Binge Alcohol Drinking+Cannabis+Cigarettes | 1.84 | (0.94, 3.58) | 0.073 | 0.94 | (0.32, 2.05) | 0.899 | 0.60 | (0.27, 2.42) | 0.369 |
| Binge Alcohol Drinking+Cannabis+Cigarettes+Nicotine vape | 2.47 | (1.35, 4.53) | 0.003*, e | 1.09 | (0.34, 3.01) | 0.877 | 1.01 | (0.37, 2.73) | 0.984 |
| Binge Alcohol Drinking+Cannabis+Nicotine vape | 4.30 | (2.44, 7.59) | <0.001*, e | 2.03 | (0.59, 5.51) | 0.212 | 3.91 | (0.60, 5.42) | 0.014* |
| Binge Alcohol Drinking+Cigarettes | 2.42 | (1.17, 5.00) | 0.017*, e | 2.44 | (0.13, 6.22) | 0.366 | 0.77 | (0.23, 3.49) | 0.704 |
| Binge Alcohol Drinking+Cigarettes+Nicotine vape | 5.54 | (2.12, 14.47) | <0.001*, e | 3.02 | (0.21, 12.49) | 0.288 | 3.04 | (0.14, 18.50) | 0.369 |
| Binge Alcohol Drinking+Nicotine vape | 1.23 | (0.64, 2.35) | 0.512 | 1.10 | (0.03, 13.58) | 0.950 | 0.37 | (0.20, 2.05) | 0.087 |
| Cannabis+Cigarettes | 1.8 | (0.87, 3.72) | 0.111 | 0.85 | (0.33, 2.06) | 0.733 | 0.65 | (0.11, 6.18) | 0.670 |
| Cannabis+Cigarettes+Nicotine vape | 2.71 | (1.34, 5.49) | 0.006*, e | 1.52 | (0.09, 3.12) | 0.641 | 0.23 | (0.12, 2.43) | 0.059 |
| Cannabis+Nicotine vape | 3.85 | (2.14, 6.93) | <0.001*, e | 1.27 | (0.26, 2.66) | 0.687 | 0.64 | (0.23, 2.98) | 0.500 |
| Cigarettes+Nicotine vape | 2.01 | (0.94, 4.32) | 0.070 | 1.02 | (0.09, 8.69) | 0.940 | 0.73 | (0.09, 8.62) | 0.791 |

"*" p-value<0.05.

PR=Prevalence Ratio.

CI=Confidence Interval.

NOTE: These are mutually exclusive categories. The reference group of the multinomial models was "No use".

[a]The sample of this model was n = 66,634. We adjusted the model by sexual identity, age group, sex, race/ethnicity, survey cycle, educational attainment, household income, urbanicity, medical insurance, employment status, and marital status. We also included an interaction term between sexual identity and sex on this model. The independent variable of this model was the interaction between sexual identity and sex.

[b]Only bisexual adults were included in this analysis. The subsample for this model was n = 5,886. We adjusted the model by age group, sex, race/ethnicity, survey cycle, educational attainment, household income, urbanicity, medical insurance, employment status, and marital status. The independent variable of this model was sex.

[c]Only LGB adults were included in this analysis. The subsample for this model was n = 8,169. We adjusted the model by sexual identity, age group, sex, race/ethnicity, survey cycle, educational attainment, household income, urbanicity, medical insurance, employment status, and marital status. We also included an interaction term between sexual identity and sex on this model. The comparisons here were bisexual female vs. gay male adults and lesbian female adults vs. gay male adults. The independent variable of this model was the interaction between sexual identity and sex.

[e] Bonferroni Correction (p < 0.005). Correction for the multiple comparisons of using multiple multinomial models (10 multinomial models in total, one per model).

Heterosexual male adults engaged more in most polysubstance combinations than heterosexual female adults, while bisexual female adults engaged more in most polysubstance combinations than bisexual male adults. However, sex was not associated with using these polysubstance combinations disproportionately among gay/lesbian adults.

We hypothesized that sex differences in the polysubstance combinations will vary by sexual identity. We also hypothesized that bisexual female adults will have higher rates among all polysubstance combinations compared to all the other groups. The study's findings partially support our hypotheses. We observed sex differences and how they vary though only among heterosexual and bisexual adults. We also found that bisexual female adults had higher rates in most of the polysubstance combinations than heterosexual female and male adults, lesbian female adults though not bisexual male adults.

Our findings support the claim that LGB groups engage in different polysubstance combinations. Therefore, substance use treatment and access to them should consider the challenges LGB groups experience and the polysubstance combinations they engage in. For example, bisexual female adults need improved access to substance use treatment, especially for binge drinking and cannabis use. Bisexual adults often have less insurance, higher unemployment, and lower

**Table 4. Summary of Survey Weighted Multinomial Models with potential combinations among bisexual female adults versus heterosexual and lesbian female adults, NSDUH 2021 and 2022.**

| Polysubstance Combinations | Bisexual Female vs. Heterosexual Female [a] | | | Bisexual Female vs. Lesbian Female [b] | | |
|---|---|---|---|---|---|---|
| | PR | 95% CI | p-value | PR | 95% CI | p-value |
| Binge Alcohol Drinking + Cannabis | 2.80 | (2.09, 3.77) | <0.001*, e | 1.84 | (1.06, 3.19) | 0.029* |
| Binge Alcohol Drinking + Cannabis + Cigarettes | 3.98 | (3.00, 5.28) | <0.001*, e | 2.34 | (1.23, 4.47) | 0.010* |
| Binge Alcohol Drinking + Cannabis + Cigarettes + Nicotine vape | 4.23 | (2.79, 6.40) | <0.001*, e | 2.27 | (1.21, 4.25) | 0.011* |
| Binge Alcohol Drinking + Cannabis + Nicotine vape | 4.24 | (3.10, 5.81) | <0.001*, e | 3.35 | (1.79, 6.28) | <0.001*, e |
| Binge Alcohol Drinking + Cigarettes | 2.12 | (1.48, 3.03) | <0.001*, e | 1.38 | (0.83, 2.29) | 0.209 |
| Binge Alcohol Drinking + Cigarettes + Nicotine vape | 3.24 | (2.16, 4.87) | <0.001*, e | 2.27 | (0.98, 5.27) | 0.058 |
| Binge Alcohol Drinking + Nicotine vape | 1.23 | (0.88, 1.73) | 0.223 | 1.22 | (0.60, 2.47) | 0.572 |
| Cannabis + Cigarettes | 3.10 | (2.20, 4.36) | <0.001*, e | 1.09 | (0.56, 2.13) | 0.797 |
| Cannabis + Cigarettes + Nicotine vape | 7.03 | (4.74, 10.44) | <0.001*, e | 1.09 | (0.48, 2.49) | 0.835 |
| Cannabis + Nicotine vape | 5.19 | (3.55, 7.60) | <0.001*, e | 2.12 | (0.97, 4.64) | 0.057 |
| Cigarettes + Nicotine vape | 3.38 | (2.23, 5.13) | <0.001*, e | 1.77 | (0.72, 4.36) | 0.218 |

"*" p-value < 0.05.

PR = Prevalence Ratio.

CI = Confidence Interval.

NOTE: These are mutually exclusive categories. The reference group of the multinomial models was "No use"

[a] Only female adults were included in this analysis. The subsample for this model was n = 37,434. We adjusted the model by sexual identity, age group, race/ethnicity, survey cycle, educational attainment, household income, urbanicity, medical insurance, employment status, and marital status. The comparisons here were bisexual female vs. heterosexual female adults and lesbian female adults vs. heterosexual female adults. The independent variable was sexual identity.

[b] Only LGB female adults were included in this analysis. The subsample for this model was n = 5,772. We adjusted the model by sexual identity, age group, race/ethnicity, survey cycle, educational attainment, household income, urbanicity, medical insurance, employment status, and marital status. The comparison here was bisexual female vs. lesbian female adults. The independent variable was sexual identity.

[e] Bonferroni Correction (p < 0.005). Correction for the multiple comparisons of using multiple multinomial models (10 multinomial models in total, one per model).

education and income than gay, lesbian, or heterosexual individuals. These factors contribute to higher social stress [42–45]. Additionally, bisexual female adults face biphobia and bi-erasure, other sources of social stress [20,21,25,26]. One way to provide access for substance use treatment while considering the potential challenges LGB adults, particularly bisexual female adults, experienced is through digital interventions. Digital interventions have shown promising results in providing access to substance use treatment to populations who have less access to healthcare settings, or experience stigma due to components of their identities or engaging in substances [46]. Furthermore, given that, on average, bisexual adults are younger than their heterosexual peers, they are more likely to have access to a smartphone, facilitating access to digital interventions for this population.

We also found that the polysubstance combinations vary by age groups; the older the age group, the lower the likelihood of using most of the assessed polysubstance combinations. Particularly, we found that ≥50 years adults engage significantly less on combinations that include nicotine vape compared to 18–29 years adults. Given that, on average, LGB adults are younger than heterosexual adults, there is a plausibility that the disproportionate use of polysubstance combinations among bisexual female adults than heterosexual female adults can be partially due to age differences. Similarly, nicotine vape might be disproportionately more used among bisexual adults than heterosexual adults due to bisexual adults being, on average, a younger group. Future work should study whether the use of these substance combinations by sexual identity and sex varies by age.

**Table 5. Summary of Survey Weighted Multinomial Models with potential combinations among heterosexual female adults versus heterosexual, bisexual and gay male adults, NSDUH 2021 and 2022.**

| Polysubstance Combinations | Heterosexual Male [a] | | | Bisexual Male [b] | | | Gay Male [c] | | |
|---|---|---|---|---|---|---|---|---|---|
| | PR | 95% CI | p-value | PR | 95% CI | p-value | PR | 95% CI | p-value |
| Binge Alcohol Drinking + Cannabis | 0.65 | (0.55, 0.77) | <0.001*, e | 0.49 | (0.33, 0.72) | <0.001*, e | 1.31 | (0.69, 2.50) | 0.411 |
| Binge Alcohol Drinking + Cannabis + Cigarettes | 0.49 | (0.40, 0.61) | <0.001*, e | 0.54 | (0.28, 1.05) | 0.069 | 0.57 | (0.25, 1.26) | 0.165 |
| Binge Alcohol Drinking + Cannabis + Cigarettes + Nicotine vape | 0.50 | (0.36, 0.71) | <0.001*, e | 0.40 | (0.22, 0.74) | 0.003*, e | 0.77 | (0.28, 2.14) | 0.616 |
| Binge Alcohol Drinking + Cannabis + Nicotine vape | 0.57 | (0.44, 0.73) | <0.001*, e | 0.23 | (0.13, 0.41) | <0.001*, e | 1.39 | (0.34, 5.70) | 0.649 |
| Binge Alcohol Drinking + Cigarettes | 0.61 | (0.52, 0.70) | <0.001*, e | 0.41 | (0.20, 0.85) | 0.017* | 0.51 | (0.22, 1.17) | 0.110 |
| Binge Alcohol Drinking + Cigarettes + Nicotine vape | 0.61 | (0.45, 0.83) | 0.001*, e | 0.18 | (0.07, 0.47) | <0.001*, e | 0.66 | (0.23, 1.91) | 0.443 |
| Binge Alcohol Drinking + Nicotine vape | 0.95 | (0.78, 1.15) | 0.594 | 0.81 | (0.42, 1.55) | 0.521 | 0.74 | (0.26, 2.13) | 0.586 |
| Cannabis + Cigarettes | 0.48 | (0.40, 0.59) | <0.001*, e | 0.56 | (0.27, 1.15) | 0.111 | 0.58 | (0.20, 1.70) | 0.316 |
| Cannabis + Cigarettes + Nicotine vape | 0.47 | (0.33, 0.67) | <0.001*, e | 0.37 | (0.18, 0.76) | 0.007* | 0.18 | (0.06, 0.53) | 0.002*, e |
| Cannabis + Nicotine vape | 0.36 | (0.25, 0.51) | <0.001*, e | 0.26 | (0.14, 0.47) | <0.001*, e | 0.34 | (0.12, 0.99) | 0.047* |
| Cigarettes + Nicotine vape | 0.76 | (0.59, 0.99) | 0.043* | 0.50 | (0.23, 1.06) | 0.069 | 0.61 | (0.13, 2.83) | 0.528 |

"*" p-value < 0.05.

PR = Prevalence Ratio.

CI = Confidence Interval.

NOTE: These are mutually exclusive categories. The reference group of the multinomial models was "No use".

[a] Only heterosexual adults were included. The subsample for this analysis was n = 58,465. We adjusted the model by age group, sex, race/ethnicity, survey cycle, educational attainment, household income, urbanicity, medical insurance, employment status, and marital status. The comparison was between heterosexual female vs. heterosexual male adults. The independent variable was sex.

[b] Only heterosexual and bisexual adults were included. The subsample for this analysis was n = 64,745. We adjusted the model by sexual identity, age group, sex, race/ethnicity, survey cycle, educational attainment, household income, urbanicity, medical insurance, employment status, and marital status. We also included an interaction term between sexual identity and sex on this model. The comparison was heterosexual female vs. bisexual male adults. The independent variable was the interaction between sexual identity and sex.

[c] Only heterosexual and gay/lesbian adults were included. The subsample for this analysis was n = 60,748. We adjusted the model by sexual identity, age group, sex, race/ethnicity, survey cycle, educational attainment, household income, urbanicity, medical insurance, employment status, and marital status. We also included an interaction term between sexual identity and sex on this model. The comparison here was heterosexual female adults vs. gay male adults. The independent variable was the interaction between sexual identity and sex.

[e] Bonferroni Correction (p < 0.005). Correction for the multiple comparisons of using multiple multinomial models (10 multinomial models in total, one per model).

Our study has multiple strengths. It is the first study to measure the past 30-day prevalence of polysubstance combinations among LGB US adults by sex in the US, focusing on the four most used substances. It also uses data from NSDUH 2021 and 2022, which are large and representative of the non-institutionalized and civilian US population, strengthening the external validity of the findings from this study. However, there are some limitations. NSDUH did not assess gender identities such as non-binary, other sexual identities like pansexual or asexual, nor sources of discrimination. In addition, NSDUH may have selection bias due to excluding people without homes and people in prison, who generally have higher rates of substance use [47,48]. There could be unmeasured confounders, such as childhood traumatic experiences, related to substance use that are more likely to occur among LGB individuals [49].

## Conclusion

Bisexual female adults tend to use more combinations of these substances compared to heterosexual and gay/lesbian male or female adults. This study also found sex differences in the combinations of these substances, and they vary

among heterosexual and bisexual adults but not among gay/lesbian adults. Future research should aim to examine why bisexual female adults engage in these polysubstance combinations to develop effective public health strategies to minimize their health risks and reduce polysubstance use among bisexual female adults. We also recommend future research that explores social, psychological, and structural drivers behind the disparities related to the disproportionate use of these polysubstance combinations. Furthermore, future work should include non-binary and transgender populations.

## Supporting information

**S1 Table. Weighted conditional percentages (%) of substances past 30 day use by sexual identity, NSDUH 2021 and NSDUH 2022 (without excluding missing data).**
(ZIP)

## Author contributions

**Conceptualization:** Luis M. Mestre.

**Data curation:** Luis M. Mestre.

**Formal analysis:** Luis M. Mestre.

**Investigation:** Luis M. Mestre, Juhan Lee.

**Methodology:** Luis M. Mestre.

**Software:** Luis M. Mestre.

**Supervision:** Krysten W. Bold.

**Validation:** Luis M. Mestre.

**Visualization:** Luis M. Mestre.

**Writing – original draft:** Luis M. Mestre, Juhan Lee, Maria A. Parker, Marney A. White, Krysten W. Bold.

**Writing – review & editing:** Luis M. Mestre, Juhan Lee, Maria A. Parker, Marney A. White, Krysten W. Bold.

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
