## [Decision Letter · Decision Letter 0]

21 Aug 2025

Dear Dr. Mestre,

Thank you for submitting your manuscript to PLOS ONE. After careful consideration, we feel that it has merit but does not fully meet PLOS ONE’s publication criteria as it currently stands. Therefore, we invite you to submit a revised version of the manuscript that addresses the points raised during the review process.

As you can see, the Reviewers have requested changes to the manuscript. Reviewer 1 was largely positive about the methodological rigour, noting the appropriate use of NSDUH data, survey-weighted regression models, and interaction effects, but recommended strengthening the robustness of the findings through sensitivity analyses and inclusion of model diagnostics. Reviewer 2, however, raised more substantial concerns, highlighting that the contribution of the study is limited as currently framed, and recommending that the analyses be expanded.

We look forward to receiving your revised manuscript.

Kind regards,

Daniel Demant, PhD

Academic Editor

PLOS ONE

Journal Requirements:

Reviewers' comments:

Reviewer's Responses to Questions

**Comments to the Author**

1. Is the manuscript technically sound, and do the data support the conclusions?

Reviewer #1: Yes

Reviewer #2: Partly

2. Has the statistical analysis been performed appropriately and rigorously?

Reviewer #1: Yes

Reviewer #2: No

3. Have the authors made all data underlying the findings in their manuscript fully available?

Reviewer #1: Yes

Reviewer #2: No

4. Is the manuscript presented in an intelligible fashion and written in standard English?

Reviewer #1: Yes

Reviewer #2: Yes

Reviewer #1: The use of data is pertinent, the authors use data from 2021 to 2022 from NSDUH a large, nationally representative, and publicly available dataset, in cross-sectional and observational study , which is appropriate for the research questions posed. They used a sample of 66, 634 subjects and identifying 8,59% of them as LGB and focuses on the four most commonly used substances: binge alcohol drinking, cannabis, cigarettes, and nicotine vape. The methods sit in a survey-weighted multinomial logistic regression models, that we consider appropriate for analyzing categorical outcomes with multiple levels, it include relevant covariates (e.g., age, race/ethnicity, education, income, etc.) and interaction terms (e.g., sexual identity × sex) to explore nuanced differences.The use of Adjusted Wald Chi-Square tests to assess sex differences is also methodologically sound. Fortunately the study uses de-identified, publicly available data and was exempt from IRB review, which is clearly stated. About discussion for support conclusions, the obtained findings tells us that bisexual female adults show the highest prevalence of polysubstance use, particularly combinations involving binge drinking and cannabis. At the same time, the sex differences in polysubstance use are significant among heterosexual and bisexual adults, but not among gay/lesbian adults, and highlights socioeconomic disparities (e.g., lower insurance coverage, income, and education) among bisexual adults, which may contribute to higher substance use.. So, the evidence of results (e.g., Tables 2–4) providing prevalence ratios with confidence intervals and p-values, many of which are statistically significant and align with the narrative conclusions. The interaction effects between sex and sexual identity are clearly presented and interpreted on the study. However the same present limitations such as the exclusion of non-binary identities, potential selection bias, and unmeasured confounders (e.g., trauma history). Concluding, the statistical analysis is methodologically sound, well-executed, and appropriate for the research questions. It supports the conclusions drawn and reflects a high level of rigor. However, while the complete-case analysis is acceptable given the low missingness (6.39%), a sensitivity analysis using imputation could further strengthen the robustness, and the manuscript could benefit from model diagnostics or goodness-of-fit measures, though these are not always standard in survey-weighted multinomial models.

Reviewer #2: In this manuscript, “Disproportionate use of polysubstance combinations varies by sexual identity among US adults”, the authors probe the prevalence and differences in the variety of combinations of substances used within the last 30 days in lesbian, gay, and bisexual individuals from the NSDUH database. Authors provide analyses that revealed that bisexual female adults had the greatest use of polysubstances, and that there were sex differences between heterosexual and bisexual adults, but not gay or lesbian adults.

The topic of understanding enhanced risk for substance use amongst populations is important, but the authors don’t fully explain why specific polysubstances may be of greater risk or why understanding which different groups of polysubstances amongst groups may be important, beyond that public health strategies should consider these. Additionally, given the authors most recent publication, the analyses presented within this manuscript seem incomplete, as they don’t take into consideration their own work and findings of differences within age groups. While large datasets are useful, and several hypotheses can be tested, the analyses presented within this manuscript are incomplete, perhaps only incremental, and should be combined for a more complete analysis.

1. While the authors state that limitations to existing studies include the mixed definitions of polysubstance use, and mixed time of sampling (30 days vs 12 mo), the current study does not do anything different than what has already been done (30 days, and a definition of polysubstance use that includes 2, 3, or 4 substances). Thus, the contribution of this study is not removing any limitations.

2. The authors have a prior publication from 2024 in which they examined the impact of age of adult on distribution patterns of polysubstance use and show differing substances endorsed, as well as patterns of polysubstance use that is dependent on age. Given those findings, it is recommended that the authors include similar analyses with this data, as the findings presented here may in fact be different for the different age groups.

3. The statistics of the analyses, and descriptions of findings are problematic. Specific instances are listed here:

a. The first problem relates to Table 1, which is described as a descriptive statistics table. However, Table 1 only presents total numbers and percentages of each demographic or substance category. There are no statistics provided, there are no ranges of variability provided. Thus, this table is not descriptive statistic, and the written description of Table 1 should not suggest that one group is more or less than the other for any characteristic, without any statistical analyses being conducted.

b. Similarly, Supplemental Table 1 shows the raw data for endorsing use of a substance, but there are no statistics to show whether there’s an actual difference between what is ranked as number 4 or 5 substance. As such, this reduces the rationale for using these substances in the analyses.

c. Additionally, in the discussion of what Supplemental Table 2 shows, the authors use terms highest and lowest, higer and lower, suggesting these are statistically significant, but there are no official statistics used for these comparisons.

d. In the third paragraph of the results section, the authors provide statistics from Weighted Wald Adjusted Chi Square Test, but don’t show the data, or indicate where the readers could see these statistics in the tables provided.

e. It is unclear whether the authors make any adjustments for multiple comparisons, given the number of multinomial models they run.

f. The title of Table 4 is incomplete.

4. There are minor grammatical errors, primarily redundancy across paragraphs, within the introduction section.

5. The authors indicate that data is available via the public URL, however, they do not provide a link to their own data, including imputed data, missing data, cases included, etc.

6. There’s a paragraph in the Measures section that includes a definition of NSDUH misuse that doesn’t belong as it is not relevant to this manuscript.

**Do you want your identity to be public for this peer review?** For information about this choice, including consent withdrawal, please see our Privacy Policy

Reviewer #1: **Yes:**  Sílvio Manuel da Rocha Brito

Reviewer #2: No

---

## [Author Response · Author response to Decision Letter 1]

6 Oct 2025

Reviewer #1: The use of data is pertinent, the authors use data from 2021 to 2022 from NSDUH a large, nationally representative, and publicly available dataset, in cross-sectional and observational study , which is appropriate for the research questions posed. They used a sample of 66, 634 subjects and identifying 8,59% of them as LGB and focuses on the four most commonly used substances: binge alcohol drinking, cannabis, cigarettes, and nicotine vape. The methods sit in a survey-weighted multinomial logistic regression models, that we consider appropriate for analyzing categorical outcomes with multiple levels, it include relevant covariates (e.g., age, race/ethnicity, education, income, etc.) and interaction terms (e.g., sexual identity × sex) to explore nuanced differences. The use of Adjusted Wald Chi-Square tests to assess sex differences is also methodologically sound. Fortunately the study uses de-identified, publicly available data and was exempt from IRB review, which is clearly stated. About discussion for support conclusions, the obtained findings tells us that bisexual female adults show the highest prevalence of polysubstance use, particularly combinations involving binge drinking and cannabis. At the same time, the sex differences in polysubstance use are significant among heterosexual and bisexual adults, but not among gay/lesbian adults, and highlights socioeconomic disparities (e.g., lower insurance coverage, income, and education) among bisexual adults, which may contribute to higher substance use.. So, the evidence of results (e.g., Tables 2–4) providing prevalence ratios with confidence intervals and p-values, many of which are statistically significant and align with the narrative conclusions. The interaction effects between sex and sexual identity are clearly presented and interpreted on the study. However the same present limitations such as the exclusion of non-binary identities, potential selection bias, and unmeasured confounders (e.g., trauma history). Concluding, the statistical analysis is methodologically sound, well-executed, and appropriate for the research questions. It supports the conclusions drawn and reflects a high level of rigor.

However, while the complete-case analysis is acceptable given the low missingness (6.39%), a sensitivity analysis using imputation could further strengthen the robustness, and the manuscript could benefit from model diagnostics or goodness-of-fit measures, though these are not always standard in survey-weighted multinomial models.

Many thanks for your feedback. We did multiple imputations for variables that were not imputed by NSDUH. In our case, sexual identity and medical insurance. We found no significant differences in the survey-weighted observed and imputed distributions of sexual identity or medical insurance among adults 18 years or older who either engaged in polysubstance use or no substances at all. We compared observed vs. imputed distribution using a survey-weighted Chi-Square Goodness of Fit Test using the Rao Scott test statistic (Supplementary Table 3a and Supplementary Table 3b).

Supplementary Table 3a – Sensitivity Analysis for sexual identity observed and imputed distribution

Sexual Identity Observed

n (%) Imputed

n (%) Resultsa

Heterosexual 59,935 (91.44) 62,321 (91.38) χ^2 = 938.45,

df = 2.86,

p-value = 0.9864

Bisexual 1,933 (2.46) 2,044 (2.48)

Gay/Lesbian 6,031 (5.50) 6,316 (5.54)

Not Sure 417 (0.60) 435 (0.60)

Total 68,316 71,1116

a Survey-weighted Rao Scott Goodness of Fit Chi-Square Test

Supplementary Table 3b – Sensitivity Analysis for having medical insurance observed and imputed distribution

Medical Insurance Observed

n (%) Imputed

n (%) Resultsa

No 6,669 (8.92) 7,077 (8.94) χ^2 = 118.29,

df = 1.00,

p-value = 0.9325

Yes 60,752 (91.08) 64,039 (91.06)

Total 67,421 71,116

a Survey-weighted Rao Scott Goodness of Fit Chi-Square Test

The model diagnostics that were available for the model diagnostics are shown in Supplementary Table 2. We only included the degrees of freedom and residual deviance that were provided by the models’ object svy_vglm. All the multinomial models were adjusted by the covariates. However, only some included the interaction between sexual identity and sex, depending on the comparison of interest (Supplementary Table 2). However, given that some models had interactions and were survey-weighted, which use survey-weighted sample sizes instead of the unweighted (i.e., observed ones), the residual deviances and degrees of freedom were expected to be large due to non-constant variance and overdispersion. We included the following statement in the methods section:

“The model diagnostics of each model are in Supplementary Table 2.”

Supplementary Table 2 – Model Diagnostics for the Multinomial Models

Multinomial Models Model Diagnostics

Including all covariates? Including interaction between sexual identity and sex? Population of interest Reference Group Sample size Degrees of freedom and residual deviance

Yes Yes General population Heterosexual Male n = 66,634 df = 732,974

Residual Deviance = 363,956,198

Yes No Female Adults Heterosexual Female n = 37,434 df = 411,774

Residual Deviance = 140,279,290

Yes Yes LGB Adults Bisexual Male

Gay Male n = 8,169 df = 89,859

Residual Deviance = 526,842,142

Yes No Gay/Lesbian Adults Gay Male n = 1,889 df = 20,779

Residual Deviance = 9,846,903

Yes No Bisexual Adults Bisexual Male n = 5,886 df = 64,746

Residual Deviance = 350,862,927

Yes No LGB female Adults Lesbian Female n = 5,772 df = 63,492

Residual Deviance = 28,600,476

Yes No Heterosexual Adults Heterosexual Male n = 58,465 df = 643,115

Residual Deviance = 265,934,111

Yes Yes Heterosexual and Bisexual Adults Bisexual Male n = 64,745 df = 712,195

Residual Deviance = 300,052,683

Yes Yes Heterosexual and Gay/Lesbian Adults Gay Male n = 60,748 df = 668,228

Residual Deviance = 277,903,266

Reviewer #2: In this manuscript, “Disproportionate use of polysubstance combinations varies by sexual identity among US adults”, the authors probe the prevalence and differences in the variety of combinations of substances used within the last 30 days in lesbian, gay, and bisexual individuals from the NSDUH database. Authors provide analyses that revealed that bisexual female adults had the greatest use of polysubstances, and that there were sex differences between heterosexual and bisexual adults, but not gay or lesbian adults. The topic of understanding enhanced risk for substance use amongst populations is important, but the authors don’t fully explain why specific polysubstances may be of greater risk or why understanding which different groups of polysubstances amongst groups may be important, beyond that public health strategies should consider these. Additionally, given the authors most recent publication, the analyses presented within this manuscript seem incomplete, as they don’t take into consideration their own work and findings of differences within age groups. While large datasets are useful, and several hypotheses can be tested, the analyses presented within this manuscript are incomplete, perhaps only incremental, and should be combined for a more complete analysis.

While the authors state that limitations to existing studies include the mixed definitions of polysubstance use, and mixed time of sampling (30 days vs 12 mo), the current study does not do anything different than what has already been done (30 days, and a definition of polysubstance use that includes 2, 3, or 4 substances). Thus, the contribution of this study is not removing any limitations.

We appreciate the reviewer’s point and agree that our study does not fully resolve the challenges of varied polysubstance definitions and timeframes in the literature. Our intent was not to eliminate these limitations, but rather to use a nationally representative sample, unlike other studies, and to be transparent in our definition of polysubstance use, the timeframe, and the contribution of polysubstance combinations. We believe our approach provides a useful step toward greater consistency in future research and possible comparison to past research in other populations and samples. We removed the paragraphs in the introduction and discussion section that could’ve caused any confusion related to varied polysubstance definitions and timeframes in the literature.

2. The authors have a prior publication from 2024 in which they examined the impact of age of adult on distribution patterns of polysubstance use and show differing substances endorsed, as well as patterns of polysubstance use that is dependent on age. Given those findings, it is recommended that the authors include similar analyses with this data, as the findings presented here may in fact be different for the different age groups.

We included the age group terms of the main multinomial model (sample size n = 66,634) in Supplementary Table 6.

We report these findings in the result section:

“In the same model, we found that adults ≥50 years and 30-49 years engaged more than 18-29 years adults in substance combinations that include cigarettes but not nicotine vape (≥50 years; Supplementary Table 6). Gay/lesbian and bisexual adults were mainly in the 18-29 years and 30-49 years groups (Table 1), indicating that some of these polysubstance combinations that were frequent among LGB adults are due to them being younger than heterosexual adults.”

We also stated in the discussion section the following:

“We also found that the polysubstance combinations vary by age groups; the older the age group, the lower the likelihood of using most of the assessed polysubstance combinations. Given that, on average, LGB adults are younger than heterosexual adults, there is a plausibility that the disproportionate use of polysubstance combinations among bisexual female adults than heterosexual female adults can be partially due to age differences. Future work should study whether the use of these substance combinations by sexual identity and sex varies by age.”

Supplementary Table 6 – Summary of Survey-Weighted Multinomial Model with potential combinations by age groups, NSDUH 2021 and 2022.

Reference Group 30-49 yearsa ≥50 yearsa

Polysubstance Combinations PR 95% CI p-value PR 95% CI p-value

Binge alcohol Drinking + Cannabis 0.74 (0.64, 0.86) <0.001*,b 0.22 (0.18, 0.28) <0.001*,b

Binge alcohol Drinking + Cannabis + Cigarettes 2.51 (2.04, 3.10) <0.001*,b 0.86 (0.67, 1.09) 0.206

Binge alcohol Drinking + Cannabis + Cigarettes + Nicotine vape 0.63 (0.44, 0.89) 0.009*,b 0.06 (0.03, 0.12) <0.001*,b

Binge alcohol Drinking + Cannabis + Nicotine vape 0.25 (0.18, 0.35) <0.001*,b 0.04 (0.02, 0.08) <0.001*,b

Binge alcohol Drinking + Cigarettes 3.50 (2.90, 4.22) <0.001*,b 1.80 (1.45, 2.22) <0.001*,b

Binge alcohol Drinking + Cigarettes + Nicotine vape 0.75 (0.54, 1.05) 0.097 0.15 (0.08, 0.26) <0.001*,b

Binge alcohol Drinking + Nicotine vape 0.44 (0.36, 0.54) <0.001*,b 0.05 (0.03, 0.08) <0.001*,b

Cannabis + Cigarettes 4.44 (3.40, 5.80) <0.001*,b 1.58 (1.13, 2.22) 0.008*,b

Cannabis + Cigarettes + Nicotine vape 1.91 (1.28, 2.86) <0.001*,b 0.16 (0.08, 0.31) <0.001*,b

Cannabis + Nicotine vape 0.54 (0.41, 0.70) <0.001*,b 0.12 (0.08, 0.18) <0.001*,b

Cigarettes + Nicotine vape 2.78 (2.05, 3.78) <0.001*,b 0.60 (0.40, 0.89) 0.011*,b

a Reference Group: 18-29 years

“*” p-value < 0.05

b Bonferroni Correction: p-value < 0.025

The model had a sample size of n = 66,634

3. The statistics of the analyses, and descriptions of findings are problematic. Specific instances are listed here:

a. The first problem relates to Table 1, which is described as a descriptive statistics table. However, Table 1 only presents total numbers and percentages of each demographic or substance category. There are no statistics provided, there are no ranges of variability provided. Thus, this table is not descriptive statistic, and the written description of Table 1 should not suggest that one group is more or less than the other for any characteristic, without any statistical analyses being conducted.

Thank you for your comment. We included in Table 1 a survey-weighted chi-square test to determine whether the covariates are directly associated with sexual identity. We also compared the estimated prevalence of each substance combination using a survey-weighted t-test between gay/lesbian and heterosexual adults, and bisexual and heterosexual adults. Furthermore, we also included the SE of the weighted prevalence (%) to provide statistics related to variability.

b. Similarly, Supplemental Table 1 shows the raw data for endorsing use of a substance, but there are no statistics to show whether there’s an actual difference between what is ranked as number 4 or 5 substance. As such, this reduces the rationale for using these substances in the analyses.

We included in Supplementary Table 1 the p-values of the survey-weighted t tests that compared the estimated prevalence of each substance between gay/lesbian and heterosexual adults, and bisexual and heterosexual adults.

We also did additional analyses using two-sample proportion tests between the use of cannabis vape and nicotine vape by sexual Identity to compare their estimated prevalence. We wrote the following in the results section:

“The most used substances in the past 30 days among gay/lesbian and bisexual adults were binge alcohol drinking, cannabis, cigarettes, and nicotine vape (Supplementary Table 1). Bisexual adults had the highest prevalence of past 30-day use of these four most common substances compared to any other sexual identity (Supplementary Table 1). We also found significant differences between the use of nicotine vape and cannabis vape by sexual identity groups; the substances ranked 4th, our threshold, and 5th, respectively, among heterosexual (χ^2(1)=1,620,361; p <0.001), gay/lesbian (χ^2(1)=130,384; p <0.001), and bisexual adults (χ^2(1)=5,288.4; p <0.001)). Therefore, our cut-off at four substances did not lose significant information from commonly used substances ranked 5th or below.”

c. Additionally, in the discussion of what Supplemental Table 2 shows, the authors use terms highest and lowest, higher and lower, suggesting these are statistically significant, but there are no official statistics used for these comparisons.

We included in Supplementary Table 2, now Supplementary Table 4, survey-weighted t-tests to compare by sex the estimated prevalence of each substance combination within sexual identity. We included the following in the results section:

“Supplementary Table 4 shows the prevalence of these common polysubstance combinations among different sexual identities and sexes. We found that for all combinations there were sex differences within the sexual identities, heterosexual, gay/lesbian, and bisexual adults (Supplementary Table 4 and Figure 1). Among heterosexual adults, male adults engaged more than female adults in most polysubstance combinations while among bisexual adults, female adults engaged more than male adults. Among gay/lesbian adults, although there were sex differences in all polysubstance combinations, they did not vary disproportionately more either among gay male or lesbian female adults.”

We adjust accordingly the first paragraph of the discussion section:

“The key findings of this study are (a) there were sex differences in polysubstance use by all sexual identities, as other studies (Ehlke et al., 2023; Flentje et al., 2024; Goodwin et al., 2022; Jun et al., 2019; Kecojevic et al., 2017; McCabe et al., 2022; Moss et al., 2014; Schauer et al., 2013; Vogel et al., 2024). (b) Bisexual female adults were at the highest risk for engaging in most of the used polysubstance combinations compared to heterosexual female adults, who were at the lowest risk, as seen in other studies (Dermody, 2018; Ehlke et al., 2023; McCabe et al., 2004, 2022; Nguyen et al., 2021; Schauer et al., 2013; Vogel et al., 2024). This pattern holds even w

---

## [Decision Letter · Decision Letter 1]

4 Nov 2025

Dear Dr. Mestre,

We look forward to receiving your revised manuscript.

Kind regards,

Daniel Demant, PhD

Academic Editor

PLOS ONE

Journal Requirements:

Reviewers' comments:

Reviewer's Responses to Questions

**Comments to the Author**

Reviewer #1: All comments have been addressed

Reviewer #2: (No Response)

2. Is the manuscript technically sound, and do the data support the conclusions?

Reviewer #1: Yes

Reviewer #2: Partly

3. Has the statistical analysis been performed appropriately and rigorously?

Reviewer #1: Yes

Reviewer #2: No

4. Have the authors made all data underlying the findings in their manuscript fully available?

Reviewer #1: Yes

Reviewer #2: Yes

5. Is the manuscript presented in an intelligible fashion and written in standard English?

Reviewer #1: Yes

Reviewer #2: Yes

Reviewer #1: This study is important because addresses poly substance use, a major public health concern linked to chronic disease and mental health risks. It focuses on sexual identity and sex differences, highlighting disparities among LGB adults compared to heterosexual adults and uses a nationally representative dataset (NSDUH) for that, improving external validity over prior localized studies. At the same time, identifies specific substance combinations (e.g., binge drinking + cannabis) most prevalent among bisexual female adults, and provides evidence to guide targeted public health strategies and treatment access for vulnerable subgroups. The research has strong points: Large, nationally representative NSDUH dataset (66k+ adults) with focus on LGB adults, highlighting bisexual women’s higher risk. At the same time demonstrate rigorous methods: survey‑weighted multinomial models, interactions, sensitivity analyses with clear reporting of limitations, public dataset, supplementary diagnostics, showing novel findings on sex differences and age‑related patterns. Finally adds to public health knowledge with policy relevance. However, present us weak points as a non‑binary/transgender groups exclusion, limiting inclusive, a limited explanation of underlying social/psychological mechanisms, and a very large residual deviances reducing the model interpretability. Finally the imputed datasets/code were not fully shared as some presentation/grammar issues either. The methods are rigorous and seems correct with the investigation using valid representative instruments. So it are reliable and reproducible, though constrained by dataset definitions and exclusions. The results results support tailoring public health strategies by sexual identity, sex, and age to address disparities.The study concludes that bisexual female adults are at the highest risk for polysubstance use. The conclusions put evidence on sex differences between heterosexual and bisexual adults, but not among on gay and lesbian adults, and the age patterns showed that younger adults drive much of the disproportionate use. The findings emphasize the need for tailored public health strategies by sexual identity, sex, and age, we talked about it on contributions. In future, authors point future researches should include non‑binary and transgender populations for broader inclusivity, should explore social, psychological, and structural drivers behind these disparities.

Reviewer #2: The authors have addressed some of the original concerns, though some existing concerns remain.

1) Inconsistent expression of Bonferroni statistics in the tables. Some tables have markers indicating bonferroni significance for tests, some have marked only some, not all, of the tests as significant following Bonferroni, some just have the marker in the legend without making changes to the table, some have the marker in one column label, some don’t reference Bonferroni at all.

2) I don’t think Bonferroni corrections have been calculated properly. There are 12 permutations of drug combinations that are being compared across the 3 sexual identities and 2 sexes. Furthermore, there were 16 individual substances that were compared across the 3 sexual identities. The authors have only stated that they’ve adjusted for the sexual identities and sexes, but even that is inconsistent in the tables as listed above.

3) Theoretical comments:

a. The very first sentence of the abstract leads with “polysubstance is a major public health concern….especially LGB female adults.” However, the authors ultimate conclusion is this is only true for bisexual female adults, not lesbian, so, the leading statement should probably be revised.

b. The paper would be easier to follow if the authors either referenced the tables when they list the specific tests they conducted, OR, if in the results, they created subsections that were titled according to which experimental question they were probing, with the appropriate results within those subsections.

c. In the discussion section, the authors lean in on “access” to substance use treatments. However, the authors should consider the design of how access is provided, and not just access, as most likely services are available, but not specifically designed for different populations and different polysubstances.

4) Grammatical comments:

a. Second paragraph first sentence of introduction should read, “…have examined….”

b. First sentence of first paragraph of discussion should read, “…as other studies have shown”

c. The last sentence of the second paragraph of the results section is awkward. Could read, “therefore we did not lose significant information by using a cut-off of 4 substances….”

d. First sentence of the discussion is not a complete sentence.

5) In the Methods, in the second paragraph of the Dependent variable (outcome) section, the first two sentences are redundant. “we ranked all substances…” and “We used the imputed variables of past 30 day substances available in NSDUH 2021 and 2022”.

6) In the same paragraph as above, why did the authors list the substances in the text as they did? It implies to the reader that there is some ranking, and the list does not match the ranking provided in S1 Table.

7) The last sentence of that same paragraph is awkward, could read instead, “Misused substances included, 13) prescription pain relievers, ….”

8) Third paragraph of that same section indicates that the combinations of substances are shown in S1 Table, but they’re not. The combinations are shown in Table 1.

9) The first paragraph for Statistical analysis states that “as recommended, we implemented NSDUH 2021 and 2022 survey weights”. For what exactly? When edited, authors can combine this sentence with the following sentence to explain the weights were divided by 2.

10) In the first paragraph for Statistical analysis, the following doesn’t make sense, “we also did survey weighted t tests to compare the estimated prevalence of (b) the substances included in the study among gay/lesbian and heterosexual adults, and bisexual and heterosexual adults”

11) In the third paragraph of statistical analysis, authors state they did 2 sample proportion test, and reference S1 Table, but that’s not where the results for the 2 sample proportion test are reported.

12) Third paragraph of results section leads with “lesbian female adults engaged more in polysubstance combinations that heterosexual male and female adults but less than gay and bisexual male adults…”, There is only 1 combination for comparison with heterosexual males that lesbian females were greater, and there was 1 combination in which lesbian females were greater than bisexual males, and a different combination in which they were greater than gay males. Thus, this leading sentence needs to be revised to more accurately reflect the results.

13) In the fourth paragraph of the results section, first sentence, why is >50 years included in the parentheses?

14) The first sentence of the first paragraph of the discussion, and the second to last sentence of that same paragraph are making opposite claims. First sentence states sex differences by all sexual identities. Second to last sentence states sex not associated in gay/lesbian adults.

15) It would help the reader if the authors revised the last sentence in the first paragraph of the discussion to include what their original hypothesis was, and what was supported vs what was not supported by the current findings.

16) The first sentence of the second paragaraph of the discussion claims that the current “findings support the claim that LGB groups….experience different challenges”. The challenges that LGB groups experience was not directly tested by this study.

17) The paragraph in the discussion section describing the age discrepancies is incomplete in that it does not describe higher combinations that include nicotine for the older age group. This should be included.

18) When discussing the limitations to the study, the authors note that NSDUH did not assess a number of things. This section could be re-written to note that in 2023, NSDUH did expand survey response choices for sexual identify.

19) First sentence of conclusion notes the most used substances among LGB adults, but this was true for all adults.

20) Table 3a does not show difference between bisexual female adult vs gay male, except for 1 combination out of 12. The description of this in the results should be revised, as the comparison is not the same as to the heterosexual male, in which there are many more.

21) S1 Table needs more descriptions for the columns of data. It is unclear what the second column (Rank) belongs to, compared to the 4th column labeled Rank, etc. There are 4 ranked columns, 4 sexual identities, only 2 p-value columns?

**Do you want your identity to be public for this peer review?** For information about this choice, including consent withdrawal, please see our Privacy Policy

Reviewer #1: **Yes:**  Sílvio Manuel da Rocha Brito

Reviewer #2: No

---

## [Author Response · Author response to Decision Letter 2]

12 Dec 2025

Reviewer #1: This study is important because addresses poly substance use, a major public health concern linked to chronic disease and mental health risks. It focuses on sexual identity and sex differences, highlighting disparities among LGB adults compared to heterosexual adults and uses a nationally representative dataset (NSDUH) for that, improving external validity over prior localized studies. At the same time, identifies specific substance combinations (e.g., binge drinking + cannabis) most prevalent among bisexual female adults, and provides evidence to guide targeted public health strategies and treatment access for vulnerable subgroups. The research has strong points: Large, nationally representative NSDUH dataset (66k+ adults) with focus on LGB adults, highlighting bisexual women’s higher risk. At the same time demonstrate rigorous methods: survey weighted multinomial models, interactions, sensitivity analyses with clear reporting of limitations, public dataset, supplementary diagnostics, showing novel findings on sex differences and age related patterns. Finally adds to public health knowledge with policy relevance. However, present us weak points as a non binary/transgender groups exclusion, limiting inclusive, a limited explanation of underlying social/psychological mechanisms, and a very large residual deviances reducing the model interpretability.

Finally the imputed datasets/code were not fully shared as some presentation/grammar issues either.

The methods are rigorous and seems correct with the investigation using valid representative instruments. So it are reliable and reproducible, though constrained by dataset definitions and exclusions. The results results support tailoring public health strategies by sexual identity, sex, and age to address disparities. The study concludes that bisexual female adults are at the highest risk for polysubstance use. The conclusions put evidence on sex differences between heterosexual and bisexual adults, but not among on gay and lesbian adults, and the age patterns showed that younger adults drive much of the disproportionate use. The findings emphasize the need for tailored public health strategies by sexual identity, sex, and age, we talked about it on contributions.

In future, authors point future researches should include non binary and transgender populations for broader inclusivity, should explore social, psychological, and structural drivers behind these disparities.

Thank you for your comment. We included the imputed datasets and code as a supplemental file. Furthermore, we included the recommended point for future research in the conclusion section: “Future research should aim to examine why bisexual female adults engage in these polysubstance combinations to develop effective public health strategies to minimize their health risks and reduce polysubstance use among bisexual female adults. We also recommend future research that explores social, psychological, and structural drivers behind the disparities in the disproportionate use of these polysubstance combinations. Furthermore, future work should include non-binary and transgender populations.”

Reviewer #2: The authors have addressed some of the original concerns, though some existing concersns remain.

1) Inconsistent expression of Bonferroni statistics in the tables. Some tables have markers indicating bonferroni significance for tests, some have marked only some, not all, of the tests as significant following Bonferroni, some just have the marker in the legend without making changes to the table, some have the marker in one column label, some don’t reference Bonferroni at all.

Please refer to comment 2)

2) I don’t think Bonferroni corrections have been calculated properly. There are 12 permutations of drug combinations that are being compared across the 3 sexual identities and 2 sexes. Furthermore, there were 16 individual substances that were compared across the 3 sexual identities. The authors have only stated that they’ve adjusted for the sexual identities and sexes, but even that is inconsistent in the tables as listed above.

Thank you for the opportunity to clarify. We included a Bonferroni Correction in our sensitivity analyses for the survey-weighted t-tests and the multinomial models. The thresholds to determine a significant p-value with Bonferroni Corrections in these t-tests were 0.004, 0.005, or 0.003; 0.05 divided by the 12 mutually exclusive categories of the polysubstance combination variable, ten multinomial models to compare female adults by sexual identity with all the other sexual identity by sex groups, 16 for the substances that were considered to operationalized the polysubstance combinations variable, respectively. We included the Bonferroni corrections of 12 mutually exclusive categories in Table 1 and S4 Table (0.05/12). We used the Bonferroni Correction of ten comparisons (0.05/10) based on the number of multinomial models used to do multiple comparisons among sexual identity by sex used in Table 2, Table 3a, Table 3b, Table 4, and S6 Table. We used a Bonferroni correction of 0.05/16 when comparing the substances used to operationalize the polysubstance combinations variable, with 12 mutually exclusive categories of the most used four substances. This Bonferroni correction is in the S1 Table.

3) Theoretical comments:

a. The very first sentence of the abstract leads with “polysubstance is a major public health concern….especially LGB female adults.” However, the authors ultimate conclusion is this is only true for bisexual female adults, not lesbian, so, the leading statement should probably be revised.

We changed the sentence as “Polysubstance use is a major public health concern affecting lesbian, gay, and bisexual (LGB) adults, especially bisexual female adults.”

b. The paper would be easier to follow if the authors either referenced the tables when they list the specific tests they conducted, OR, if in the results, they created subsections that were titled according to which experimental question they were probing, with the appropriate results within those subsections.

We added two subsections within the result section, as recommended by the reviewer. “Sex Differences in the Polysubstance Combinations vary by Sexual Identity” and “Bisexual Female Adults have Higher Rates among Polysubstance Combinations compared to Heterosexual Female, Heterosexual Male, Gay Male, and Lesbian Female adults.”

c. In the discussion section, the authors lean in on “access” to substance use treatments. However, the authors should consider the design of how access is provided, and not just access, as most likely services are available, but not specifically designed for different populations and different polysubstances.

We added the following: “One way to provide access for substance use treatment while considering the potential challenges LGB adults, particularly bisexual female adults, experienced is through digital interventions. Digital interventions have shown promising results in providing access to substance use treatment to populations who have less access to healthcare settings, or experience stigma due to components of their identities or engaging in substances [46]. Furthermore, given that, on average, bisexual adults are younger than their heterosexual peers, they are more likely to have access to a smartphone, facilitating access to digital interventions for this population.”

4) Grammatical comments:

Thanks for the catches. We fixed these grammar issues.

a. Second paragraph first sentence of introduction should read, “…have examined….”

We changed it to “… have examined…”

b. First sentence of first paragraph of discussion should read, “…as other studies have shown”

We changed the first paragraph of the discussion section so it read “…as other studies have shown.”

c. The last sentence of the second paragraph of the results section is awkward. Could read, “therefore we did not lose significant information by using a cut-off of 4 substances….”

We included the reviewer’s recommendation.

d. First sentence of the discussion is not a complete sentence.

We changed the first sentence to ensure is a complete sentence as follows:

“The key findings of this study are the following: (a) there were sex differences in polysubstance use among bisexual and heterosexual adults, as other studies have shown [9,10,12,14,28–31,33]; (b) bisexual female adults were at the highest risk for engaging in most of the used polysubstance combinations compared to heterosexual female adults, who were at the lowest risk, as seen in other studies [8,9,12,14,27,32,33].”

5) In the Methods, in the second paragraph of the Dependent variable (outcome) section, the first two sentences are redundant. “we ranked all substances…” and “We used the imputed variables of past 30 day substances available in NSDUH 2021 and 2022”.

We removed the sentence “We ranked all substances…”.

6) In the same paragraph as above, why did the authors list the substances in the text as they did? It implies to the reader that there is some ranking, and the list does not match the ranking provided in S1 Table.

To avoid confusion, we removed the listing and just stated the substances.

7) The last sentence of that same paragraph is awkward, could read instead, “Misused substances included, 13) prescription pain relievers, ….”

We fixed the sentence as recommended.

8) Third paragraph of that same section indicates that the combinations of substances are shown in S1 Table, but they’re not. The combinations are shown in Table 1.

We changed it to Table 1.

9) The first paragraph for Statistical analysis states that “as recommended, we implemented NSDUH 2021 and 2022 survey weights”. For what exactly? When edited, authors can combine this sentence with the following sentence to explain the weights were divided by 2.

We combined the sentences to avoid redundancy. The sentence now reads as: “We implemented NSDUH 2021 and 2022 survey weights; we divided the survey weights by two, the number of survey cycles included in the study.”

10) In the first paragraph for Statistical analysis, the following doesn’t make sense, “we also did survey weighted t tests to compare the estimated prevalence of (b) the substances included in the study among gay/lesbian and heterosexual adults, and bisexual and heterosexual adults”

We changed it to the following: “(b) the substances included in the study to determine the polysubstance combinations”

11) In the third paragraph of statistical analysis, authors state they did 2 sample proportion test, and reference S1 Table, but that’s not where the results for the 2 sample proportion test are reported.

We removed “S1 Table” as correctly stated by the reviewer.

12) Third paragraph of results section leads with “lesbian female adults engaged more in polysubstance combinations that heterosexual male and female adults but less than gay and bisexual male adults…”, There is only 1 combination for comparison with heterosexual males that lesbian females were greater, and there was 1 combination in which lesbian females were greater than bisexual males, and a different combination in which they were greater than gay males. Thus, this leading sentence needs to be revised to more accurately reflect the results.

We changed the sentence as follows: “Lesbian female adults engaged more in polysubstance combinations than heterosexual female adults (Table 2). Lesbian female adults only engaged more in one combination than heterosexual male adults, though they engaged less in one combination compared to bisexual and gay male adults.”

13) In the fourth paragraph of the results section, first sentence, why is >50 years included in the parentheses?

We removed the parentheses and the “>50” to avoid confusion.

14) The first sentence of the first paragraph of the discussion, and the second to last sentence of that same paragraph are making opposite claims. First sentence states sex differences by all sexual identities. Second to last sentence states sex not associated in gay/lesbian adults.

We specified the following in the first sentence to have consistency: “The key findings of this study are the following: (a) there were sex differences in polysubstance use among bisexual and heterosexual adults, as other studies have shown.”

15) It would help the reader if the authors revised the last sentence in the first paragraph of the discussion to include what their original hypothesis was, and what was supported vs what was not supported by the current findings.

We added a small paragraph to bolster the last sentence and include the hypotheses.

“We hypothesized that sex differences in the polysubstance combinations will vary by sexual identity. We also hypothesized that bisexual female adults will have higher rates among all polysubstance combinations compared to all the other groups. The study’s findings partially support our hypotheses. We observed sex differences and how they vary, though only among heterosexual and gay/lesbian adults. We also found that bisexual female adults had higher rates in most of the polysubstance combinations than heterosexual female and male adults, lesbian female adults, though not bisexual male adults.”

16) The first sentence of the second paragaraph of the discussion claims that the current “findings support the claim that LGB groups….experience different challenges”. The challenges that LGB groups experience was not directly tested by this study.

We removed the “experience different challenges” of the sentence.

17) The paragraph in the discussion section describing the age discrepancies is incomplete in that it does not describe higher combinations that include nicotine for the older age group. This should be included.

We changed the paragraph as the following to include information about nicotine vape and older ages.

“We also found that the polysubstance combinations vary by age groups; the older the age group, the lower the likelihood of using most of the assessed polysubstance combinations. Particularly, we found that ≥50 years adults engage significantly less on combinations that include nicotine vape compared to 18-29 years adults. Given that, on average, LGB adults are younger than heterosexual adults, there is a plausibility that the disproportionate use of polysubstance combinations among bisexual female adults than heterosexual female adults can be partially due to age differences. Similarly, nicotine vape might be disproportionately more used among bisexual adults than heterosexual adults due to bisexual adults being, on average, a younger group. Future work should study whether the use of these substance combinations by sexual identity and sex varies by age.”

18) When discussing the limitations to the study, the authors note that NSDUH did not assess a number of things. This section could be re-written to note that in 2023, NSDUH did expand survey response choices for sexual identify.

NSDUH 2023 includes an additional response survey for sexual identity. However, those were “I use a different term” and “I am not sure about my sexual identity.” None of these options assessed specific sexual identities.

19) First sentence of conclusion notes the most used substances among LGB adults, but this was true for all adults.

We removed the first sentence.

20) Table 3a does not show difference between bisexual female adult vs gay male, except for 1 combination out of 12. The description of this in the results should be revised, as the comparison is not the same as to the heterosexual male, in which there are many more.

We removed the gay male group from our statement in the results section to avoid any confusion.

21) S1 Table needs more descriptions for the columns of data. It is unclear what the second column (Rank) belongs to, compared to the 4th column labeled Rank, etc. There are 4 ranked columns, 4 sexual identities, only 2 p-value columns?

There are only two p-value columns. One for gay/lesbian adul

---

## [Editor Report · Decision Letter 2]

21 Dec 2025

Disproportionate use of polysubstance combinations varies by sexual identity among US adults

PONE-D-25-25612R2

Dear Dr. Mestre,

We’re pleased to inform you that your manuscript has been judged scientifically suitable for publication and will be formally accepted for publication once it meets all outstanding technical requirements.

Kind regards,

Daniel Demant, PhD

Academic Editor

PLOS One
---

## [Editor Report · Acceptance letter]

PONE-D-25-25612R2

PLOS One

Dear Dr. Mestre,

I'm pleased to inform you that your manuscript has been deemed suitable for publication in PLOS One. Congratulations! Your manuscript is now being handed over to our production team.

Kind regards,

on behalf of

Associate Professor Daniel Demant

Academic Editor

PLOS One